# Dissolution of Molybdenum in Hydrogen Peroxide: A Thermodynamic, Kinetic and Microscopic Study of a Green Process for ^99m^Tc Production

**DOI:** 10.3390/molecules28052090

**Published:** 2023-02-23

**Authors:** Flavio Cicconi, Alberto Ubaldini, Angela Fiore, Antonietta Rizzo, Sebastiano Cataldo, Pietro Agostini, Antonino Pietropaolo, Stefano Salvi, Vincenzo Cuzzola

**Affiliations:** 1ENEA, C.R. Brasimone, 40032 Camugnano, Italy; 2ENEA, Via Martiri di Monte Sole 4, 40129 Bologna, Italy; 3ENEA, S.S.7 “Appia” Km 706, 72100 Brindisi, Italy; 4ENEA, Via E. Fermi 45, 00044 Frascati, Italy

**Keywords:** molybdenum, dissolution thermodynamics and kinetics, hydrogen peroxide, neutron source, fusion

## Abstract

^99m^Tc-based radiopharmaceuticals are the most commonly used medical radioactive tracers in nuclear medicine for diagnostic imaging. Due to the expected global shortage of ^99^Mo, the parent radionuclide from which ^99m^Tc is produced, new production methods should be developed. The SORGENTINA-RF (SRF) project aims at developing a prototypical medium-intensity D-T 14-MeV fusion neutron source specifically designed for production of medical radioisotopes with a focus on ^99^Mo. The scope of this work was to develop an efficient, cost-effective and green procedure for dissolution of solid molybdenum in hydrogen peroxide solutions compatible for ^99m^Tc production via the SRF neutron source. The dissolution process was extensively studied for two different target geometries: pellets and powder. The first showed better characteristics and properties for the dissolution procedure, and up to 100 g of pellets were successfully dissolved in 250–280 min. The dissolution mechanism on the pellets was investigated by means of scanning electron microscopy and energy-dispersive X-ray spectroscopy. After the procedure, sodium molybdate crystals were characterized via X-ray diffraction, Raman and infrared spectroscopy and the high purity of the compound was established by means of inductively coupled plasma mass spectroscopy. The study confirmed the feasibility of the procedure for production of ^99m^Tc in SRF as it is very cost-effective, with minimal consumption of peroxide and controlled low temperature.

## 1. Introduction

Technetium-99m (^99m^Tc) is a metastable nuclear isomer of technetium-99 that is used in tens of millions of medical diagnostic procedures annually [1], making it the most commonly used medical radioisotope in the world [2]. Radiopharmaceuticals based on ^99m^Tc are used mainly in single-photon emission computed tomography (SPECT), and, for this reason, this isotope is of great importance in nuclear medicine [3]. None of the Tc isotopes are stable, the one with the longest half-life being (t_1/2_) ^98^Tc, equal to 4.2 million years [4]. This means that this element can only be found in traces in nature, and, hence, all isotopes must be artificially produced by nuclear reactions, in particular ^99m^Tc (t_1/2_ = 6.0 h), which is normally derived from its transient equilibrium parent, ^99^Mo (t_1/2_ = 66 h) [1,2].

^99^Mo decays by emitting a beta particle (an electron). About 88% of the decays produce ^99m^Tc, which subsequently decays to the ground state, ^99^Tc, by emitting a gamma ray. About 12% of the nuclear decays produce ^99^Tc directly. In turn, it decays to stable ruthenium-99 (^99^Ru) after emitting a beta particle with a half-life of 211.1 thousand years [3,5].

A technetium-99m generator based on molybdenum-99 is commercially available. The generator is easy to transport and use, which are some of the reasons why it is so widely used in hospitals all over the world [6].

^99^Mo can be produced following different methods, for example, using accelerated charged-particle beams (α-particle capture via ^96^Zr (α, n)^99^Mo reaction or fast proton interaction with ^100^Mo via ^100^Mo (p,2n) ^99m^Tc reaction) or according nuclear reactions in which fast neutrons are involved: neutron photo-production in ^100^Mo via ^100^Mo (γ, n)^99^Mo reaction or fast neutron interaction in ^100^Mo via ^100^Mo(n,2n)^99^Mo inelastic reaction [1,7,8,9,10,11].

However, despite these methods, at present, ^99^Mo is almost exclusively obtained from fission of ^235^U-containing targets, irradiated in a small number of research nuclear fission reactors in the world [2,12,13].

This fact can lead to a series of non-negligible issues. A global shortage of ^99^Mo is a risk, and it happened during the late 2000s because of frequent shutdown due to extended maintenance periods of the main reactors for ^99^Mo production, namely the Chalk River National Research Universal (NRU) nuclear fission reactor in Canada and the High Flux Reactor (HFR) in the Netherlands. They are capable of meeting about two-thirds of ^99^Mo world demand [14,15]. These events highlighted vulnerabilities in the supply chain of medical radionuclides that relies on nuclear fission reactors.

Indeed, as a fission product, ^99^Mo is produced together with many other isotopes of various elements, from which it must be purified [13]. This requires development and implementation of specific and complex radiochemical processes to separate the isotope of interest from all the rest, which, therefore, constitutes a waste product. Therefore, there is a general problem of waste management, and the threat of nuclear proliferation must always be considered [12,15]. It should be kept in mind that ^99^Mo accounts for only about 6% of uranium fission products [13,16]. Large volumes of hazardous chemicals, including strong acids, are required for this purpose [13], and, for this reason, ^99^Mo production cannot be considered environmentally friendly.

In this context, the SORGENTINA-RF (SRF) project aims at developing a prototypical medium-intensity D-T 14-MeV fusion neutron source mostly dedicated to production of medical radioisotopes, with a special focus on ^99^Mo. Indeed, the fusion neutron route is very interesting for a series of reasons, but the lack of a very intense 14 MeV neutron source is a limitation factor for ^99^Mo production. SRF will be a prototype plant to assess this production route [1,17].

Theoretically, an alternative route can be followed in order to produce ^99^Mo, relying on use of 14 MeV neutrons from a deuterium–tritium fusion reaction:D + T → ^4^He + n + 17.6 MeV
and on inelastic channel ^100^Mo(n,2n)^99^Mo [1,8,9,18].

The idea is to exploit the neutrons generated by the fusion process to irradiate a metal target made up of metallic natural molybdenum, where ^100^Mo has 10% abundance. The accelerator will operate with deuterons and tritons that will be implanted onto a titanium layer a few microns thick where they interact, in turn producing a neutron field, the main component being that from the D-T reaction mentioned above [19].

First, calculations and projections from a dedicated study indicate that the end of irradiation (EoI) activity of ^99^Mo is in the range 2–5 Ci after 24 h continuous irradiation starting from an initial mass of about 10 kg. This yield is more than enough for the daily needs of ^99m^Tc of the entire Emilia Romagna, an Italian region, and could be improved by using samples enriched in ^100^Mo and higher potencies. The SRF project is, therefore, extremely promising due to the numerous advantages it can offer compared to more traditional methods [1].

In contrast to the traditional production methods, with the SRF method, there are no radiochemical purification issues. The main challenge becomes finding an effective and ecologically acceptable method to transform metallic molybdenum into a stock solution of sodium molybdate, Na_2_MO_4_, which is used for feeding the Mo/Tc generators [13].

In the case of the SRF prototype, once irradiated, the molybdenum target is transferred into shielded hot cells for dissolution and radiochemical processing.

The stock solution could be prepared by dissolving the metal target using concentrated strong acids or aqua regia, and, in the past, this has usually been completed [13]. However, a greener and more ecologically acceptable approach is to use less aggressive reagents and in the smallest amount possible to achieve the same result. Micrometric Mo powders react vigorously with hydrogen peroxide, even if diluted. In this case, the only by-products are water vapor and oxygen, which is a clear advantage in terms of the sustainability of the process. If coarse pieces of a few centimeters are used, the overall process has slower chemical kinetics (it should also be kept in mind that the molybdenum dissolution process, in the context of SRF, must be completed in a maximum time equal to that necessary for irradiation of the target, i.e., 24 h [20]). Nevertheless, it has been reported that a large amount of highly concentrated hydrogen peroxide can be effective also in the case of samples with the shape of disks [21,22]. In particular, the authors of [22] used hydrogen peroxide for direct production of ^99m^Tc from ^100^Mo by cyclotrons.

The aim of this work is, therefore, to find the most efficient conditions possible to optimize the dissolution process using hydrogen peroxide in a suitable time. Furthermore, a description of the chemical path followed to arrive at formation of the stock solution is also presented.

The chemical behavior of metallic molybdenum towards hydrogen peroxide is studied, from a thermodynamic and kinetic point of view, by means of scanning electron microscopy, infrared and Raman spectroscopy, X-ray diffractometry, pH, temperature and conductivity measurements. Use of metal not exposed to the neutron beam is acceptable in this context because it can be assumed that the physicochemical properties are not substantially modified by irradiation. Two different target geometries have been investigated: pellets and powder. Therefore, once the most suitable conditions for dissolution of the non-irradiated target have been determined, they can also be applied to the target irradiated by SRF.

## 2. Experimental Sections

### 2.1. Materials and Methods

The molybdenum pellets were supplied by Luoyang Combat Tungsten & Molybdenum Material Co., Ltd. (Luoyang, China). They are shaped as small cylinders, approximately 1.7 mm long and 1.5 mm in diameter. Figure 1 shows the optical micrograph images of them in order to show the aspect and morphology. All pieces are very similar and just small variations in volume can be observed. Their surface is usually dark, probably because of the presence of a very thin layer of molybdenum oxide on the surface. Purity of the samples has been analyzed by means of ICP-mass spectrometry (see Section 3.3).

The dissolution process has been studied, performing many experiments with hydrogen peroxide solutions whose concentration ranged from to 3% to 35% *w*/*w* and using a mass of metallic molybdenum between 0.5 and 100 g. Different hydrogen peroxide solutions ranging from 30 to 40% *w*/*w* were obtained from Carlo Erba Reagents (Milan, Italy), Titolchimica (Pontecchio Polesine, Italy) and Sigma-Aldrich (St. Louis, MO, USA), and they were diluted, if necessary, to the desired concentration for the experiments.

An external ice bath was used in order to control the temperature because the large exothermicity of the reaction can lead to a very fast increase of the system temperature.

Highly concentrated hydrogen peroxide is unstable and decomposes over time. For this reason, fresh batches were used and the solutions were kept in freezer at −18 °C.

The dissolution experiments were performed under fume hood. For each experiment, the procedure has been the following: the hydrogen peroxide solution is prepared with the desired concentration from a stock solution. The solution is then transferred in the flask with an ice-cooled water bath, the molybdenum pellets are weighed with analytical scale and then are added slowly to the hydrogen peroxide solution under magnetic stirring. A circuit with a peristaltic pump connected to an inlet of cold water ensures circulation of the water bath. The low temperature of the solution delays the initial states of the reaction and prevents formation of foam. Furthermore, if the solution is poured directly over the molybdenum pellets, the reaction starts too vigorously and a considerable fraction of the peroxides are consumed via self-decomposition and subsequent formation of a huge amount of foam can overflow from the flask, leading to failure of the experiment. The procedure described above is, therefore, much cleaner and milder and avoids drastic and uncontrollable changes in the temperature reaction.

### 2.2. Characterization Methods

In the present work, Raman spectra of the compounds were acquired by a BWTEK i-Raman plus spectrometer equipped with a 785 nm laser to stimulate Raman scattering, which is measured in the range 100–3500 cm^−1^ with a spectral resolution of 3.5 cm^−1^. The measurement parameters, acquisition time, number of repetitions and laser energy have been selected for each sample in order to maximize the signal to noise ratio. For each spectrum, reference acquisition with the same parameters was previously carried out to subtract the instrumental background. Infrared spectra were acquired with a Thermo Fisher Scientific Nicolet 6700 FT-IR with SMART iTX ATR System (Thermo Fischer Scientific, Waltham, MA, USA) in the mid-, far- and near-IR ranges from 4000 to 400 cm^−1^).

On selected materials, X-ray powder diffraction (XRPD) investigations were performed in order to determine the crystalline phases present using a Philips X’Pert PRO 3040/60 diffractometer operating at 40 kV, 40 mA, Bragg–Brentano geometry, equipped with a Cu Kα source (1.54178 Å), Ni filtered and a curved graphite monochromator. PANalytical High Score software was utilized for data elaboration.

Characterization of the samples, morphology and composition has been performed by scanning electron microscopy (SEM-FEI Inspect-S) coupled with energy-dispersive X-ray spectroscopy (EDX-EDAX Genesis).

The pH and EC of the solutions during the dissolution process were measured using a Crison Basic 20 pH-meter and a Crison Basic 30 EC-Meter (Crison, HachLange, Spain).

A triple quadrupole inductively coupled plasma mass spectrometer (QQQ-ICP-MS, 8800 model, Agilent Technologies, Santa Clara, CA, USA) equipped with two quadrupoles, one (Q1) before and one (Q2) after the octopole reaction system (ORS3), installed in a dedicated Clean Room ISO Class 6, (ISO 14644-1 Clean room) with controlled pressure, temperature and humidity was used for trace analysis of residual metals in the molybdenum.

The instrument was calibrated with reference samples from a multi-element stock standard solution containing 13 elements (100 mg/L, each in 7% HNO_3_
*v*/*v*, P/N: CCS-6) supplied by Inorganic Ventures (Christiansburg, VA, USA) as an external calibration. The multi-element standard includes 13 elements that are mainly transition metals that are used in the metallurgic industries, namely V, Cr, Mn, Fe, Co, Ni, Cu, Zn, Ag, Cd, Hg, Tl, Pb. Hg, Tl, Pb, which were also added in the analysis for their toxicological concerns even though they are not commonly present in metallic matrixes.

All data collection and analysis were performed in ICP-MS Tandem MS/MS Helium Mode using Mass Hunter software. Quality control standards (QCs) were added in the batch of analysis in order to control the accuracy of the analytical method. The analyte recoveries were within 90% of the true values.

LODs and LOQs were estimated, respectively, as three and ten times the standard deviation (σ) of 10 consecutive measurements of the reagent blanks according to EURACHEM recommendation [23].

The collision cell in Helium Mode MS (MS/MS) configuration ensures removal of any spectroscopic interferences are caused by atomic or molecular ions that have the same mass-to-charge as analytes of interest. Regarding those isobaric interferences for the case of molybdenum, some of its oxide’s ions can lead to overestimation of Cadmium isotopes; in fact, ^95^Mo^16^O^+^ is isobaric to ^111^Cd and ^98^Mo^16^O^+^ is isobaric to ^114^Cd [24].

All the reagents used in the experiment were analytical reagent grade. Trace SELECT^®^ grade 69% HNO_3_ and 37% HCl and ultra-pure grade 30–32% H_2_O_2_ were acquired from Sigma-Aldrich (St. Louis, MO, USA) and Carlo Erba Reagents (Milan, Italy), respectively.

High-purity de-ionized water (resistivity 18.2 MOhm cm^−1^) was obtained from a Milli-Q Advantage A10 water purification system (Millipore, Bedford, MA, USA).

## 3. Discussion

### 3.1. Studies on the Dissolutions Process

Molybdenum has many industrial applications because of its excellent physical properties: for instance, high resistance versus corrosion [25], much slower than steel [26], low linear expansion coefficient, relatively high thermal and electrical conductivity and excellent mechanical characteristics, even at very high temperature, such as high tensile strength and stiffness [27]. Therefore, many of its chemical physical features have been deeply studied.

It is a transition metal with moderate reactivity, which is strongly determined by specific surface, presence of impurities and traces of oxide and, in general, by presence of defects on the surface as they can act as catalytic starting points for any reaction or transformation [28].

This metal belongs to the chromium group and has a rich chemistry because it has oxidation states ranging from −II to +VI and coordination numbers from 0 to 8 [29]. It can form several oxides [30], among which the most important and commonly observed are MoO_3_ (that has, in turn, some different polymorphs [31]) and MoO_2_, but at least seven other oxides with molybdenum oxidation state comprised between 4 and 6 exist, for instance, Mo_8_O_23_ [32] and Mo_17_O_47_ [33]. The oxide of trivalent Mo ion Mo_2_O_3_ also exists; indeed, some hydrated compounds have been reported, such as MoO_3_–2H_2_O or MoO_3_–H_2_O. In aqueous solutions, higher oxidation states are more relevant, and, in general, compounds of Mo (VI) are more soluble, so the goal of the dissolution process is to form a solution containing molybdate ions (MoO_4_^2−^) [29,34].

This metal also has several commonalities with the chemistry of tungsten (W), even if it reacts more easily with strong inorganic acids and oxidants, such as hydrogen peroxide.

In the case of reaction with hydrogen peroxide, schematically, the process can be described according to the following formal steps:Oxidation of metallic molybdenum from the surface toward the center to form a mixture of low- and intermediate-valence molybdenum oxide.Strong oxidation of these ions to the highest valence state by reaction with concentrated hydrogen peroxide to form some soluble oxyhydroxides and peroxides of molybdenum, such MoO_2_(O_2_)_2_^2−^ or MoO_2_HO(O_2_)^2−^ [34].Reaction of these species with NaOH to form Na_2_MoO_4_.

The stoichiometry of the first two steps cannot be perfectly defined, so an overall formal reaction can be written as such:Mo(s) + 3 H_2_O_2_ (l) + 2 NaOH (l) → Na_2_MoO_4_ (l) + 4 H_2_O(1)

Figure 2 shows the stages of dissolution: on the left, the initial stage with ice bath and the peristaltic pump for a temperature-controlled reaction; in the center, the orange solution at the end of the dissolution with the intense orange color due to molybdenum-peroxo complexes species; on the right, the transparent solution of sodium molybdate after addition of sodium hydroxide.

It has been previously studied by Tkac et al. [35] that the dissolution rates and foam produced by the reaction are highly dependent on the specific brand of peroxide solutions using sintered Mo disk. This is presumably caused by the different peroxide stabilizers used in the final product. The most common stabilizers are metal-based (such as Sn or Al) or phosphate-based; they can both affect the dissolution reaction, and the metals can produce a catalytic effect, increasing the dissolution rate, for example, or the stabilizer simply reduces formation of free peroxide radicals, leading to a different reaction rate. They also reported that dissolution rate is dependent on the specific surface of the sintered disk, as expected. In the preliminary phases of this study, experiments have been conducted regarding how the different peroxides brands can affect reaction rate with a specific ice-cooled bath and peristaltic pump set-up. The reaction rate experiments were not affected by the different peroxide type: Titolchimica, Carlo Erba or Sigma-Aldrich. This is probably because of the controlled temperature that disadvantages collateral reactions, such as parasitic self-decomposition of peroxide itself. For these reasons, for subsequent experiments, a single brand has been chosen in order to enhance repeatability and only the Sigma-Aldrich “For Trace Analysis” grade has been used.

Electrochemical methods have also been investigated. Cieszykowska et al. [36] studied electrochemical dissolution of sintered molybdenum disks in the scale of about 10 g using additional peroxide solution and potassium hydroxide as electrolyte with current density of 365 mA/cm^2^ and temperature of 55 °C.

In general, the dissolution processes of solids in liquids are faster if the reaction surface is greater. For this reason, molybdenum powders would be preferable and they react with the peroxide much faster than macroscopic pieces due to the fact that, for the same mass, the surface is bigger. Therefore, very large pieces are not suitable for achieving this purpose, which is to have rapid formation of the stock solution, because the process would take too long or require much greater quantities of peroxide.

Use of molybdenum metal powders makes the reaction very fast, to the point that, as has been observed during the present research, a great deal of foam can form during the process even when the reaction is ice-cooled. Associated with this phenomenon, it has also been observed that even small quantities of powders, when they are quickly brought into contact with the liquid, lead to a large increase in system temperature, up to about 380 K. Furthermore, the foam could easily fill the chemical reactor and eventually overflow, with obvious safety problems, and can remove the smallest and lightest metallic particles from the reaction environment, making it difficult to achieve complete dissolution. The strong increase in temperature leads to decomposition of hydrogen peroxide before it has completely reacted, and, consequently, a large excess of solution will be necessary.

However, powders are not suitable within the SRF project for one fundamental reason: once irradiated, the molybdenum target becomes strongly radioactive and cannot be manipulated or transported by a human operator. This means that all the phases following irradiation must be automated. The target will be automatically transferred by means of a pneumatic system from the bunker where the neutron irradiation is carried out to the hot cells for radiochemical processing. For this reason, small particles or powders cannot be used because they can clog the filters, remain attached to the walls of the pipes and be easily and dangerously dispersed into the facility. The ideal geometry has, therefore, been identified as pellets or beads of millimeter size.

Several tests were carried out using different concentrations of hydrogen peroxide ranging from 3% to 30% *w*/*w*. Taking constant all the other parameters, the time required for the dissolution process decreases almost linearly with peroxide concentration, suggesting that the control steps are phenomena occurring at metallic pellets surface. It is interesting to note that, during this rapid reaction, the color of the solution can change long before reaching the final orange–yellow one because different oxyanions and molybdenum complexes are formed. Even though, from a formal point of view, the reaction is quite simple, in reality, the chemical path leading to formation of the desired products is rather complex [34,37].

It has been observed experimentally in this study that temperature does not increase immediately when the reactants come into contact; there is an initial induction time before which the temperature remains practically constant, and, immediately after, it increases dramatically. This induction time is generally of few seconds and depends on peroxide concentration (as well as temperature and other experimental factors), suggesting that the reaction acts as if it were self-catalytic, as can be seen in Figure 3. It shows the trend of the time required for appearance of the first bubbles and its inverse as a function of concentration of hydrogen peroxide solution, when the metallic pieces are thrown into the liquid. In this case, just a few milligrams of Mo and a few milliliters of peroxide were used.

The trend of the inverse time clearly suggests that the concentration of the peroxide is the fundamental factor and the kinetics of the process are likely controlled by phenomena at solid/liquid interface.

A possible explanation for this phenomenon can be found in the fact that the molybdenum pieces are covered with a very thin layer of MoO_2_ that acts as a passivation layer, and that is why they appear dark and somehow dusty on the surface. Indeed, because of this layer, generally, in aqueous conditions, molybdenum exhibits considerably lower corrosion rates than many metals, such as iron [34,38]. This oxide forms easily because of favorable redox potentials in these two half-reactions (0.152 V and 1.23 V, respectively) [34,39]:Mo + 2 H_2_O → MoO_2_ + 4 H^+^ + 4 e^−^
O_2_ + 4 H^+^ + 4 e^−^ → 2H_2_O

The reaction path must necessarily involve this layer as well. Tyurin [40] suggested that, in alkaline, neutral and weakly acidic solutions, this passive layer is stable.

However, this oxide can be oxidized to the +VI state at higher potentials and low pH, as shown in Figure 4 regarding the E–pH diagram for pure molybdenum [40,41,42]. Hydrogen peroxide itself is a weak acid, but, as the reaction proceeds, the pH decreases so that the conditions favor dissolution.

Thus, molybdenum dissolves at more positive potentials as an oxyanion of molybdic acid depending upon pH, i.e., as molybdic acid H_2_MoO_4_ at low pH, as hydrogen molybdate anion between pH 3 and 5 and as molybdate anion above about pH 5–6. Some more complex equilibria can exist and other polyoxomolybdates may form [43,44].

**Figure 4 molecules-28-02090-f004:**
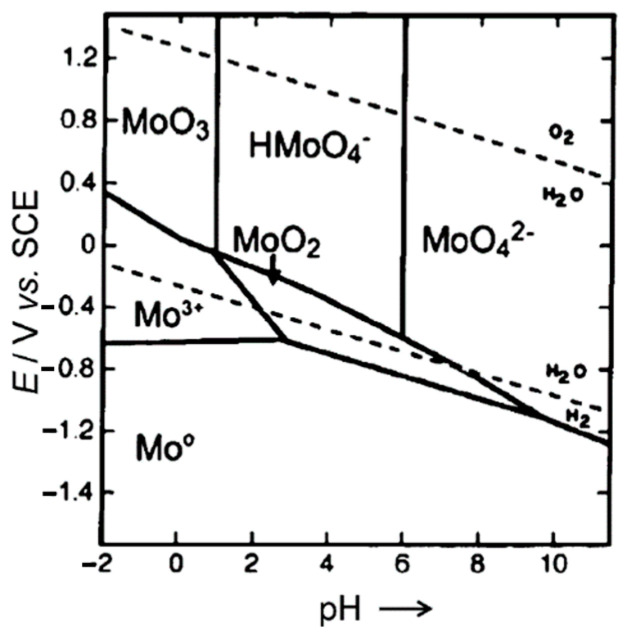
E-pH diagram for molybdenum taken from [41].

As the oxidation process proceeds, according to Badawy [45], other oxides form, such as Mo_2_O_5_, with lower protective capacity and formation of a thick oxidation product. This could offer a clue as to whether induction time is observed. As long as the passivating layer of MoO_2_ exists, the oxidation reaction of the metal cannot take place; on the contrary, the more this passivation layer is destroyed, the more the underlying metal reacts quickly. Furthermore, as the reaction proceeds, pH decreases because the main chemical species with which the molybdenum passes into solution is anion HMoO_4_^−^, making the reaction even faster. For electroneutrality of a system, pH obviously decreases.

Figure 5a shows the trend of pH and electroconductivity as a function of time during the dissolution process of 20 g of molybdenum using a 30% peroxide solution with external ice bath, and Figure 5b shows the trend of the temperature during the experiment. The mass variation (defined as (Mi − Mf)/Mi × 100, where Mi and Mf are the initial and final mass, respectively) of the undissolved metal was also measured in other experiments and shown in the same figure. The final yield of this test after 300 min was over 99%, meaning that dissolution of the metal is complete under these conditions.

It is important to note that, due to the external ice bath, in the first half of the experiment, pH, electrical conductivity of the solution and temperature are almost constant and the mass of metal remains almost constant, indicating that the reaction is very slow at this stage. However, after about 200 min, all these parameters change drastically. The pH of the solution decreases by about 0.5 units and the conductivity increases more than ten times. The temperature, which, until then, had remained roughly stable, increases more than 303 K in a few minutes despite the ice bath, and the mass of the samples begins to decrease dramatically and the inflection points coincide with a drastic increase in temperature.

**Figure 5 molecules-28-02090-f005:**
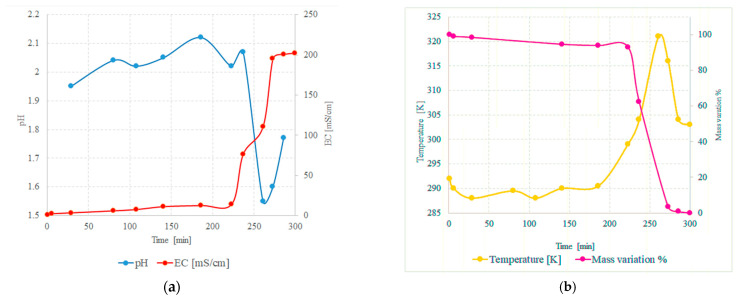
(**a**) pH and electroconductivity of solution during the dissolution process and (**b**) temperature and mass variation dependence on time during the same process.

The onset of temperature rise coincides with the moment when pH and conductivity begin to vary significantly. The pH fluctuations must be evaluated even though the measurement conditions were not optimal due to the increase in viscosity and the different complexes present in the solution. It is possible that the final rise in pH may be linked to a variation in the anionic species present in the solution and their equilibria. For example, monomeric molybdate ion can easily form polymolybdate anions [43,44] (in most cases, with general formula Mo_x_O_3x+12_^−^ or HMo_x_O_3x+1_^−^) and possibly peroxopolymolybdates, in particular when the concentration, as in this case, is high [46]. Indeed, under particularly acidic conditions, anion MoO_4_^2−^ can be subject to some protonation reaction, which leads to formation of species such as MoO_3_(OH)^−^ and MoO_2_(OH)_2_(OH_2_)_2_. All these chemical equilibria can lead to an increase in pH [46]. After complete dissolution of the molybdenum, an orange-colored limpid solution is obtained.

All these observations are in excellent agreement with the proposed model for molybdenum dissolution. In fact, by using the ice bath, it is possible to slow down the first step of the process so that it is easier to observe the drastic variation in the reaction after induction time. As long as the MoO_2_ passivation layer is almost intact [42,46,47], the process is very slow, whereas, after its removal, it becomes extremely fast and self-catalytic. At this point, the temperature spikes and the electrical conductivity of the system increases dramatically, meaning that the concentration of ions in solution increased significantly. In fact, it is also observed that the mass of the sample begins to decrease only in correspondence with these phenomena and the pH also decreases, in agreement with the hypothesis that, initially, the main molybdenum species in the solution is anion HMoO_4_^−^.

These observations also suggest that, in order to optimize the reaction rate for larger batches, the system temperature should be controlled; it should not be too low, i.e., below about 300 K; otherwise, the process is too slow, and it should not be too high as this would lead to competitive decomposition of the peroxide before complete dissolution, thus requiring larger amounts of hydrogen peroxide, and this is not suitable for an environmentally cleaner process. The optimum process temperature appears to be around 320 K.

It is also observed, in excellent agreement with this chemical path model, that the molybdenum pellets appear with a metallic-silvery surface and no longer black just before sudden changes take place.

Finally, solid sodium hydroxide was added to the intensely orange acid solution until complete discoloration, obtaining a final pH of 14. During the reaction, the NaOH reacts with the peroxides according to the reaction 2 NaOH + H_2_O_2_ → Na_2_O_2_ + 2 H_2_O. The reaction is exothermic and, therefore, favors removal of the residual hydrogen peroxide.

### 3.2. Microscopic Characterization

In order to investigate the dissolution mechanism more deeply, some pieces of molybdenum were observed under electron microscope at different reaction times, keeping all the other parameters constant.

Figure 6 shows the temporal evolution of their surface and the appearance of corrosion signs.

In this figure are shown low-magnification secondary electron-SEM pictures of Mo pellets dissolved in peroxide solution at different reaction times, basal (left column) and lateral (right column) overviews of: Figure 6a,b starting Mo pellets; Figure 6c,d Mo pellets after 30 minutes; Figure 6e,f after 60 minutes and Figure 6g,h after 120 minutes of reaction; all scale bars are 200 µm).

Figure 6a,b reports a representative starting Mo pellet showing the basal face without any cavities and holes. After dissolution in peroxide solution, the surface becomes defective and the pieces become smaller on average. Specifically, pristine Mo pellets clearly present dissolution and corrosion signs. The basal faces display presence of numerous cavities and lateral overviews provide direct observation of corrosion lines focused on the edges of the cylinder (Figure 6d,f,h). Upon increasing reaction time, extent of corrosion lines increases such that, after 120 min of reaction time, some pellets achieve a barrel shape (Figure 6h).

Obviously, as the atoms are oxidized to the hexavalent state and pass into solution, the mass of the metallic piece decreases.

However, the size reduction is not uniform in all directions. Although the pieces are not all exactly alike, this observation is absolutely general. In other words, the diameter decreases in percentage much more slowly than length, indicating that, even at the microscopic level, the process follows a particular path. Loss of mass is, therefore, not a simple and progressive release of matter from the external surface of the beads towards the liquid phase until they are completely consumed.

The effects upon corrosion lines of reaction time are reported in Table 1 and Table 2; the trend is similar to behavior already observed with regard to pellet evolution: the diameter is almost constant, whereas the length rapidly decreases.

EDX-SEM investigations (not reported here) were carried out on the entire surface of our four samples to provide compositional information, with a particular focus on understanding the presence of oxygen. Slight oxidation was reported both on flat and lateral surfaces, with uniform distribution of Mo and oxygen content. However, comparing the analysis of the four samples, no significant difference was detected, both in terms of content and distribution of two elements.

Corrosion of a solid surface by liquids can be a very complex phenomenon from a physicochemical point of view and there are many different microscopic mechanisms that can be involved [48].

There are some cases where corrosion occurs in a uniform manner on the whole surface, often called uniform or general corrosion, referring to a process that proceeds at approximately the same rate over the whole exposed surface. However, often, corrosion preferentially starts where the surface energy is greater, i.e., in correspondence with cracks, defects, grain boundaries and presence of impurities [49]. Localized corrosion leads to pitting, which greatly enhances dissolution process kinetics [50]. Regardless of the specific mechanism, localized corrosion begins at specific sites, and, once the process has begun, these initial sites become larger over time and involve the entire surface.

Corrosion can be intragranular or intergranular [51], being the second a form of attack where the boundaries of crystallites of the material are more susceptible to the dissolution process than their insides. Intergranular corrosion is a form of localized corrosion, where the corrosion takes place in a quite narrow path preferentially along the grain boundaries in the metal structure.

It seems realistic to imagine that these small pieces are manufactured by hot extrusion and that, therefore, the grains align themselves preferentially along their axis. Then it can be imagined that they are mechanically cut to the desired size. Their shape can be approximated to cylinders. Often, traces of carbon used as a lubricant for extrusion can remain on the side surfaces, but not on the planar ones. These molybdenum pieces are of high purity, so the planar surfaces are quite uniform. Furthermore, it is possible that the MoO_2_ layer on the side surface may be thicker because it can be formed during a heat treatment.

Based on the SEM images, it appears that molybdenum dissolution by peroxide is a very complex combination of uniform and localized corrosion. The fact that the sample pieces progressively shorten while their diameter remains roughly the same indicates uniform corrosion, which mainly affects flat surfaces. The metallic pieces progressively lose mass from them, without any particular corrosion points appearing in the first stages at least. On the contrary, the lateral surface shows evident signs of very localized corrosion, with lines that become progressively more evident. In this case, the corrosion is more of the intergranular type since a combination of the two processes cannot be completely ruled out at least initially; once the process has taken hold, it becomes prevalent.

It is, therefore, possible to suppose a different mechanism on the different sides: uniform corrosion on the planar faces and localized corrosion on the lateral surface.

Dissolution on the planar faces starts at the beginning uniformly over the entire surface and on both sides in a similar manner. However very soon, there is formation of some deep corrosion points on these surfaces, with morphology that is quite reminiscent of pitting (Figure 7c). These craters have a conical appearance in the sense that, the deeper they are, the smaller their radius. These corrosion marks, especially the larger ones, are often separated by thin walls that meet at angles of about 120°. In general, pitting is a form of extremely localized corrosion that leads to random creation of small holes and often is due to the de-passivation of a small area. The corrosion rate is, among other factors, determined by the size of the surface that is corroded: the greater, the faster the process. In this case, as the signs of corrosion appear new, surface attackable by the peroxide is created, making the dissolution faster and faster.

Since this does not occur on the lateral surface, this can offer an explanation for the fact that the loss of mass occurs mainly from the basal surface rather than from the lateral surface.

Actually, the later surface does not participate to the dissolution process in the very early stages; only its extremities are involved and, in these positions, the signs of corrosion are very evident. After some time, some dissolution lines propagate from these initial points and run along the lateral surface from both sides until they arrive to touch and cover the entire lateral surface (Figure 7a,b). At this point, corrosion proceeds quickly and also inwards. The possible microscopic dissolution mechanism that has been discussed is depicted schematically in Figure 8.

### 3.3. Spectroscopic Characterization

As described in the previous sections, the final aim of the radiochemistry task within the SRF project is to efficiently and quickly prepare a stock solution to obtain ^99m^Tc. However, one of the indirect ways to prove the quality of the method used in this work and demonstrate the purity of the solution is to obtain crystals of sodium molybdate, Na_2_MoO_4_, and characterize them spectroscopically. Furthermore, this compound is an interesting material on its own due to its various properties and is worth studying as it stands. For example, it is a promising material for optoelectronics [52].

After preparation of the sodium molybdate solution, the crystals have been slowly evaporated, dried in a laboratory oven and analyzed.

Sodium molybdate at room temperature and pressure crystallizes in the spinel type structure, with Fd-3 m symmetry and cell parameter a = 9.10888 Å [53,54,55]. It is possible to observe high purity and excellent quality of the sample obtained in this way since almost all the peaks of the diffractogram can be uniquely attributed only to the pure Na_2_MoO_4_ compound and there are only few spurious signals that are due to a very small quantity of sodium carbonate, Na_2_CO_3_, which, in turn, very likely derives from small quantity of impurities in the NaOH and can, therefore, be avoided using this reagent with better quality and higher purity. This also means that the method, when perfectly controlled, is very effective in obtaining the stock solution for feeding the Mo/Tc generators.

Figure 9 below shows the XRD pattern of a sample crystallized from this solution by evaporating the liquid, and Figure 10 and Figure 11 show Raman spectrum and IR spectrum of the same sample, respectively.

All the Raman bands and signals in IR spectrum can be uniquely attributed only to the pure anhydrous Na_2_MoO_4_ compound and no spurious peaks can be detected.

The crystalline structure consists of isolated MoO_4_ tetrahedra and the crystals have eight formula units per unit cell [36,37,38]. In the crystal, MoO_4_ groups occupy Td sites and Na ions occupy D3d sites. Vibrational modes of these tetrahedra are the main features that can be observed in both types of spectra and the position of observed bands depends on the characteristic vibrations of the O–Mo–O bonds and MoO_4_ blocks, even if collective modes of the whole crystalline lattice can have a role. According to standard group theory, the irreducible representation Γ admits 39 optical modes that are [53,54,55].
Γ = A_1g_ + E_g_ + F_1_g + 3F_2g_ + 2A_2u_ + 2E_u_ + 4F_1u_ + 2F_2u_

Selection rules enable A_1g_, E_g_, and F_2g_ modes to be Raman-active due to symmetric stretching (υ1), asymmetric stretching (υ3), symmetric bending (δ2) and asymmetric bending (δ4) of the MoO_4_ units. The measured bands match what has been reported in the literature: 891 cm^−1^, 808 cm^−1^, 380 cm^−1^ and 302 cm^−1^. In addition to these bands, it is possible to note a further signal at about 115 cm^−1^ that can be assigned to translational collective mode of the crystalline lattice [53]. The assignments are reported in Table 3. In the case of IR spectrum instead, the band at about 550 cm^−1^ can been attributed to the bending vibration of the MoO_4_^2−^ tetrahedron, while the signals in the range 650–800 cm^−1^ are due to the stretching vibration of the tetrahedra observed also; in particular, the very intense band at about 830 cm^−1^ is due to O–Mo–O stretching [54,55].

It is important to note the absence of signals in around 3000 cm^−1^, which is the region due to hydrogen–oxygen stretching. This indicates significant absence of water, of crystallization nature or absorbed, in the sample.

**Table 3 molecules-28-02090-t003:** Assignments of the Raman bands of sodium molybdate.

Wavenumber (cm^−1^)	Attribution
115	Collective Mode (MoO4)
302	δ (Mo–O)
380	δ (Mo–O)
808	ν_as_ (Mo–O)
891	ν_s_ (Mo–O)

The purity of starting materials and final product have been verified by ICP-mass spectrometry and resulted in very high values, as can been seen in Table 4.

The molybdenum samples were prepared in a clean laboratory for sample treatment and preparation of the ICP-MS analysis with a microwave digestion system, Speedwave Four model (Berghof, Germany), equipped with temperature and pressure control for digestion vessels made of PTFE.

Both the molybdenum pellets, which are the chosen geometry for this study, and the molybdenum powder, which was used in the preliminary phases of this research, were analyzed and compared to the purified crystal of sodium molybdate at the end of the process. In this way, it has been possible to evaluate any unexpected contamination from the beginning to the end of the process.

The values of the analyzed elements were all well beyond the limits of the certificate analysis of the products (<500 ppm for both powder and pellets), and the requirements for trace metal analysis are met, confirming the high purity of the starting material. Importantly, secondary elements concentration did not increase (with the only exceptions of lead and cadmium, whose concentrations remain, in any case, far below the safety limits); on the contrary, it decreased, meaning that the dissolution/recrystallization process increases purity.

RSD for all measurements were below 5%.

## 4. Conclusions

In this experimental study, a green, mild and cost-effective molybdenum dissolution procedure has been investigated. Up to 100 g of solid molybdenum have been successfully dissolved in less than 6 h, demonstrating the feasibility of the process for preparation of the solution used in Mo99/Tcm99 generators. The developed method is rapid and involves mild reaction conditions, minimal reagents consumption and quantitative yields. Furthermore, reaction temperature is easily controlled with ice bath, and, therefore, risks and hazards are minimized.

Chemical and kinetic measurements have also been performed for parameters optimization, such as solution pH and EC, hydrogen peroxide concentration and induction time during the dissolution process.

Sodium molybdate crystals have been successively recrystallized and characterized by means of X-ray diffractometry, Raman spectroscopy, infrared spectroscopy and inductively coupled plasma mass spectroscopy, showing the very high purity of the compound and the extremely low contamination of other metals (<80 ppb).

We further studied the mechanism of dissolution and corrosion of molybdenum pellet via scanning electron microscopy and EDX analysis in order to better understand the dissolution chemistry of the specific solid molybdenum geometry and optimize future larger-scale experiments with specifically designed reaction vessels.

We can, therefore, conclude that the method fits the standards needed for radiochemical processing of irradiated molybdenum by 14 MeV fusion neutron source SORGENTINA-RF.

## Figures and Tables

**Figure 1 molecules-28-02090-f001:**
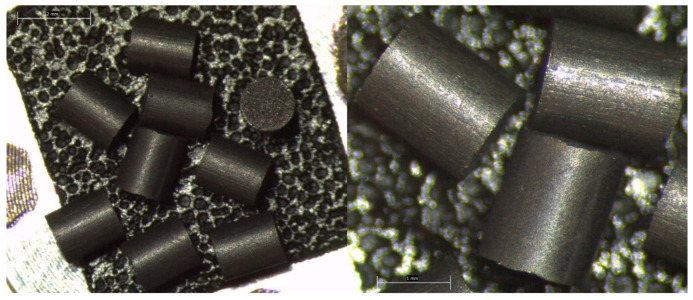
Optical micrograph of the metallic molybdenum pieces at different magnifications.

**Figure 2 molecules-28-02090-f002:**
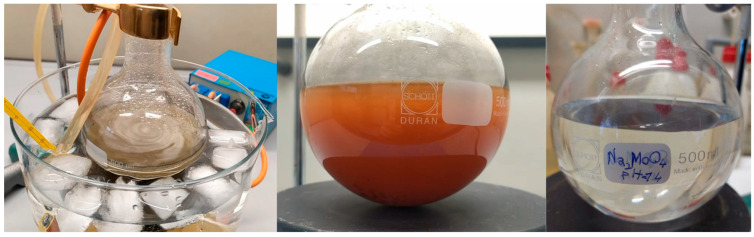
Images of the dissolution process of the molybdenum pieces.

**Figure 3 molecules-28-02090-f003:**
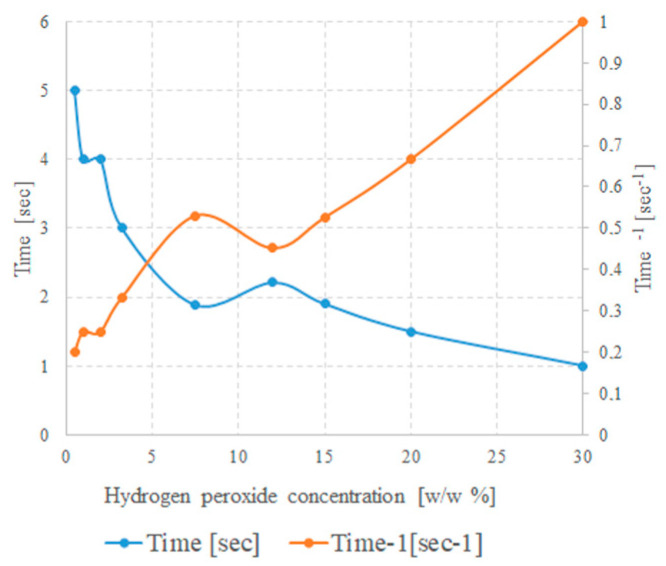
Dependence of time of appearance of the first bubbles at surface of metallic pieces and its inverse as a function of concentration.

**Figure 6 molecules-28-02090-f006:**
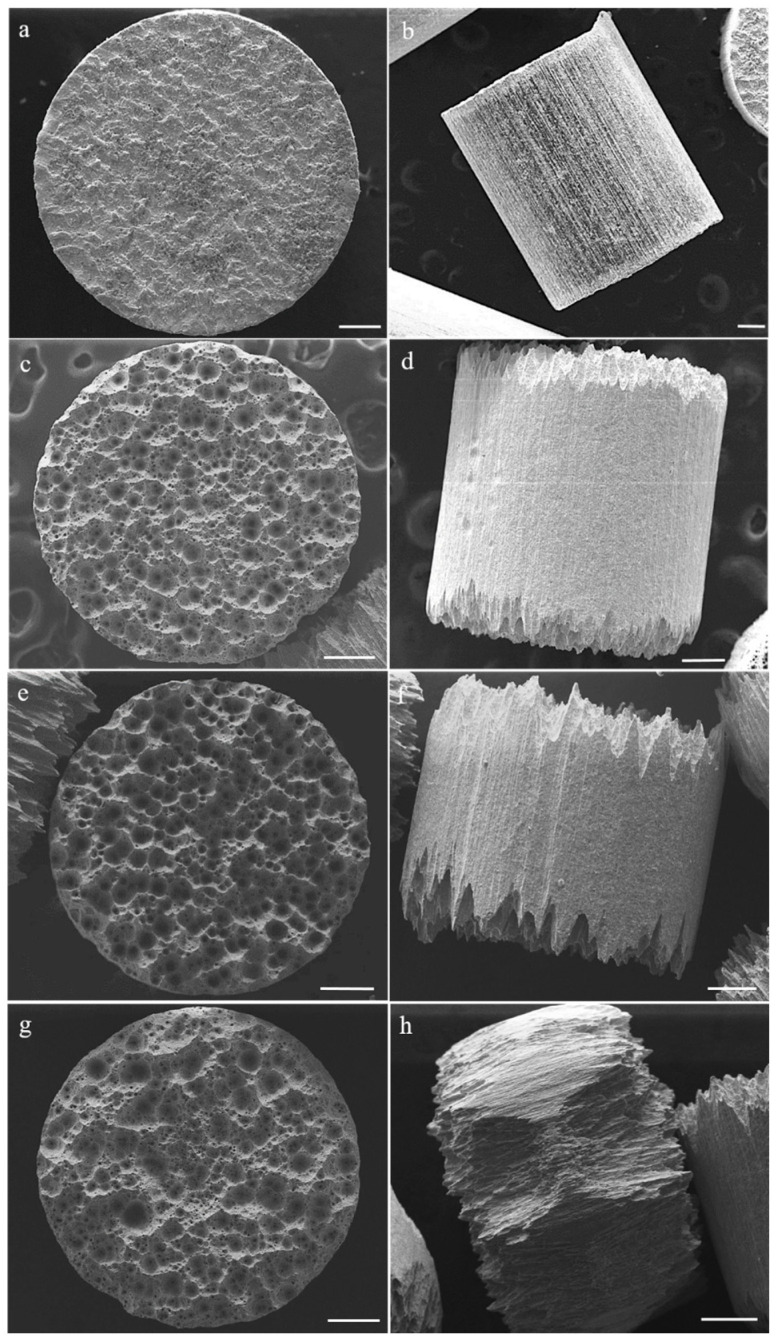
SEM images of some molybdenum pieces from top row to bottom row: 0 min ((**a**) basal view, (**b**) lateral view) 30 min ((**c**) basal view, (**d**) lateral view), 60 min ((**e**) basal view, (**f**) lateral view), 120 min ((**g**) basal view, (**h**) lateral view).

**Figure 7 molecules-28-02090-f007:**
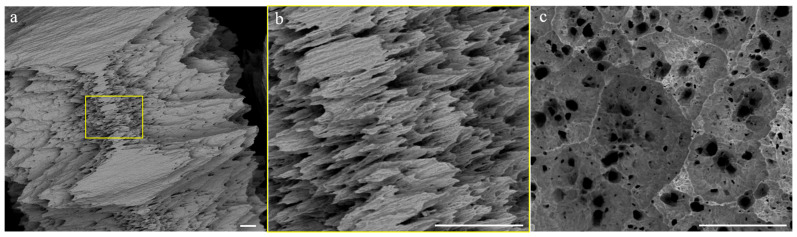
Backscattered SEM images of Mo pellet as obtained after 120 min of reaction time. Lateral (**a**,**b**) and basal (**c**) overview of Mo pellet presenting evident localized corrosion. (**b**) Zoom of Figure 7a showing the corrosion lines run along the lateral surface until reaching the middle of the barrel-shaped pellet. All scale bars are of 50 µm.

**Figure 8 molecules-28-02090-f008:**
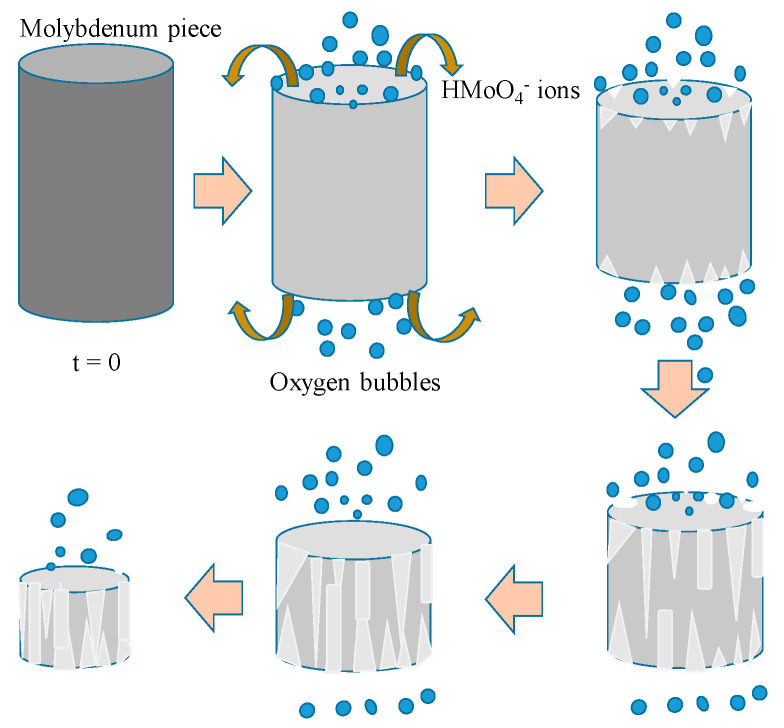
A model representation of the proposed microscopic mechanism.

**Figure 9 molecules-28-02090-f009:**
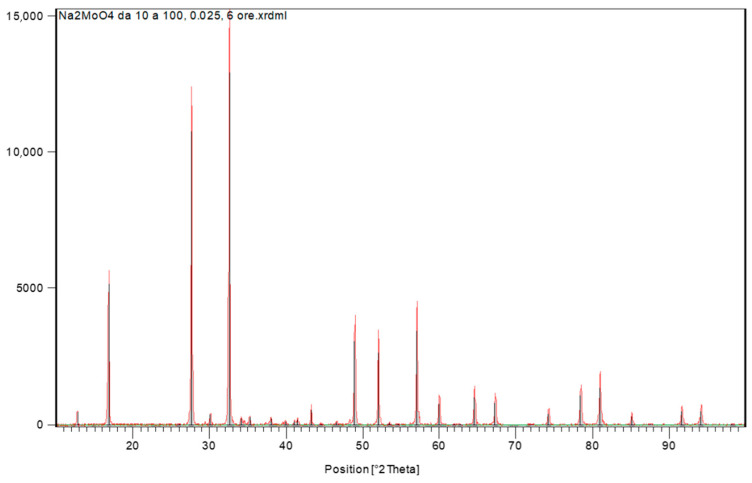
XRD pattern of sodium molybdate.

**Figure 10 molecules-28-02090-f010:**
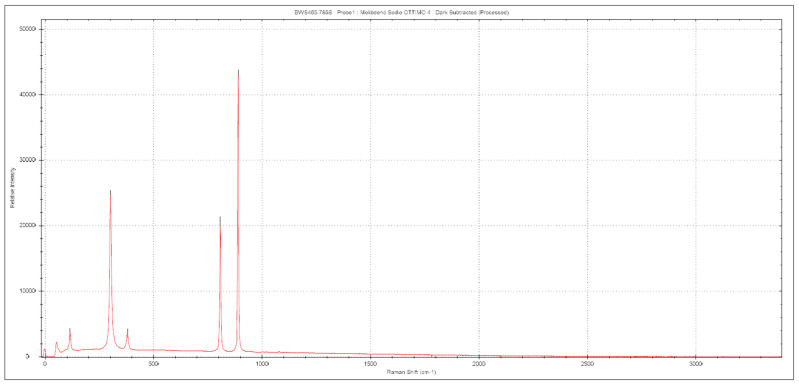
Raman spectrum of sodium molybdate.

**Figure 11 molecules-28-02090-f011:**
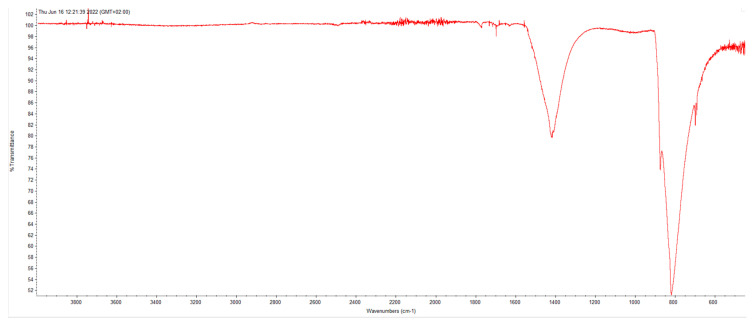
IR spectrum of the same sample.

**Table 1 molecules-28-02090-t001:** Mean dimensions of the diameter and the lengths of the pellets at various dissolution times in peroxide solution 30%.

Reaction Time	Mean Pellet Diameter (µm)	Mean Pellet Length (µm)
0 min	1490 ± 10	1710 ± 10
30 min	1253 ± 10	1113 ± 10
60 min	1265 ± 10	935 ± 10
120 min	1210 ± 10	626 ± 10

**Table 2 molecules-28-02090-t002:** Mean dimensions of the diameter and the lengths of the pellets and maximum values for diameter and length of corrosion lines as obtained by SEM image analysis at various dissolution times in peroxide solution 30%.

Reaction Time	Mean Corrosion Lines Diameter (µm)	Mean Corrosion Lines Length (µm)	Max Value of Corrosion Lines Diameter (µm)	Max Value Corrosion Lines Length (µm)
30 min	1253 ± 10	1113 ± 10	118	271
60 min	1265 ± 10	935 ± 10	108	408
120 min	1210 ± 10	626 ± 10	126	626

**Table 4 molecules-28-02090-t004:** Impurities concentration in the initial molybdenum pieces and in the final product (* below LOQ).

Element	Concentration (ppb)		
	Pellets (LCTMM Co.)	Powder (Merck, DE)	Purified Sodium Molybdate
V	0.361	1.061	0.252
Cr	23.72	12.18	8.793
Mn	2.998	1.184	1.620
Fe	161.2	75.78	76.81
Co	8.181	2.478	9.574
Ni	16.99	18.51	9.995
Cu	2.204	1.397	0.901
Zn	4.130	2.446	1.242
Ag	0 *	0 *	0 *
Cd	71.28	53.69	77.63
Hg	0.449	0.452	0 *
Tl	0 *	0 *	0 *
Pb	1.653	1.026	8.556

## Data Availability

Not applicable.

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
