# Peer review of "Dissolution of Molybdenum in Hydrogen Peroxide: A Thermodynamic, Kinetic and Microscopic Study of a Green Process for 99mTc Production"

_molecules, 2023, doi:10.3390/molecules28052090_

Round 1

Reviewer 1 Report (Previous Reviewer 1)

The new version of the paper is more clear, the changes in the text make it more neat and easier to read. Previos suggestions of correction have been attended and new and valuable references had been added.

Reviewer 2 Report (New Reviewer)

The manuscript represents a detailed experimental study of a procedure for the dissolution of molybdenum in hydrogen peroxide.

The motivation of this study is clearly stated in the introduction. The authors note the current and future problems and difficulties in the worldwide production and supply chain of 99Mo, a radionuclide which decays to 99mTc, the most used radionuclide in nuclear medicine imaging. The dissolution of molybdenum is essential in order to obtain the sodium molybdate required for the 99Mo/99mTc generators, thus the results generated by this study are relevant regardless of the methods of obtaining 99Mo.

The authors present a series of chemical and kinetic measurements in order to optimize the dissolution process such as sample geometry, reaction temperature, pH, EC, hydrogen peroxide concentration and induction time.

Furthermore, SEM and EDX analysis of the Mo pellets at different times during the dissolution process provide insight into the corrosion mechanism on the samples.

The resulting sodium molybdate solution was recrystallized and structurally characterized by means of X-Ray diffraction, Raman and IR spectroscopies. All the diffractogram peaks where attributed to pure Na2MoO4, and all IR and Raman bands were successfully assigned. Sample purity was analyzed by inductively coupled plasma mass spectroscopy revealing an extremely low contamination of impurities.

The authors propose an optimized, cost effective procedure for the dissolution of molybdenum in hydrogen peroxide. It will be interesting to see how well this procedure scales when using large quantities of irradiated molybdenum in order to address the current problems in the production and supply chain.

This manuscript is a resubmission of an earlier submission. The following is a list of the peer review reports and author responses from that submission.

Round 1

Reviewer 1 Report

The works presents an interesting approach to the Molibdenum global shortage. The paper is clearly presented, the experiments are correctly done. Minor corrections should be done, they are highlighted in the attached revised version.

Author Response

Dear reviewer,

We want first of all to thank you for all your comments, absolutely useful and sharp.

Thanks to them, we can make a much better manuscript. We accept all the comments you made because they are very precise.

Best regards

Reviewer 2 Report

The manuscript is a detailed physical-chemistry study of the dissolution of larger pieces of metallic  molybdenum in hydrogen peroxide solution. It draws heavily on long-known principles of inorganic chemistry, and it cites known Redox-pH phase state diagrams of molybdenum in aqueous solution.

The paper " discovers" some well known facts:

-Reaction rate is proportional to surface versus mass ratio.

-Reaction rate is proportional to concentration of H2O2.

-H2O2 is not stable at high ( boiling) temperatures.

-The oxidation of Mo is,- once initiated, extothermic, and cooling is necessary to prevent loss of H2O2 due to boiling, and,  - secondary oxidation of the Mo metal.

-A stable orange crystalline compound can be formed (sodium molybdate) by addition of sodium hydroxide and careful evaporation to dryness.

None of this is hardly new. However the detailed studies and the precise measurements of  temperature and reaction rate as well as the "modern methods" of x-ray diffraction,  Raman and ICP-MS in the characterization of the reaction products is probable above the level found anywhere in the old literature. 

This said, the basic objections to this paper are:

1) The justification of the paper is based on the fast neutron production of Mo-99 from kilogram quantities of metallic, natural molybdenum. True enough, such a scheme will require dissolution of the molybdenum, but NOT on the time scale of the irradiation length (line 124 etc), but on the time scale of half-life of Mo-99.  This leaves much more time for the chemical processing.

2) Even at a speculative high radionuclide yield of 2-5 Ci/24 hours, the molar activity of Mo-99 from such a production route is VERY low as compared with what is needed for any modern Mo/99/Tc-99m generator system. A typical hospital generator at 50 GBq Mo-99 would then be needed to be loaded with  hundreds of grams of molybedenum.  This makes the generator itself less feasible and unpractical, and the resulting radioactive concentration of Tc-99m probably very low. Unless this problem is somehow solved ( or a solution explained) the rapid dissolution of kg's of molybdenum is not really relevant in this context.

3) While it is correct that finely powdered radioactive molybdenum will not be easy to handle in dry, open state, it can (and is) routinely and safely handled when the powder is encapsulated during irradiation and transport. More so, water soluble chemical compounds of molybdenum could be directly irradiated, making the difficult metal dissolution step unnecessary.

4) The technology of rapid dissolution of radioactive metallic molybdenum (Mo-100) targets of 100-500 mg size has already been proven. The so called "Instant Technetium" cyclotron route is already using this technique, where cyclotron targets are dissolved and Tc-99m extracted on a time scale << 6 hours.  Some manufacturers prefer the dry distillation, but the wet dissolution route is also well developed.  A proper referencing to this technology ( pioneered by TRIUMF in Canada) should be made , at least.

These fundamental objections apart, it does not help that the manuscript itself is talkative and self-repeating in several places ( see as example lines 170-174 as compared to lines 174-177 and also lines 289-297 being exactly identical to lines 302-313) The "explanation" of the "Sorgentina RF" concept is also "explained in several places.

The reference to  table 3 in line 429 should be reference to table 2 !

The ordinate axis in fig.8b is missing ( and how can mass disappear from a chemical reaction?)    

The SORGENTINA RF is described as being both 300 W (line 84) and 250 kW ( line 243).  That is confusing. The description of the titanium target is a mistake. It is the H-3 that is implanted into the titanium target. This target is then bombarded with deuterons.  

It is not justified why the all the elaborate analytical methods of XRD, Raman and ICP-MS are needed to establish the simple ( and visually obvious, from crystal shape and color)  fact that sodium molybdate has been formed. As long as there is no practical generator concept for this kind of Mo-99, the fine chemical details are unnecessary.   

The section 3.2 about the the TEM microscope imaging of the dissolution process  leads to a speculative ( and lengthy ) explanation of the physical chemistry behind the fact that ends are dissolved at higher rate than cylinder surfaces . To this reviewer, it looks more like an artefact of the manufacturing process of the beads. They have most likely been hot drawn under high pressure and then "chopped" into short sections, leaving the grain structure aligned and closed on the cylinder surface and fractured on the ends. 

However, in recognition of the large piece of experimental work done, some nice experimental/analytical details and the large effort that has been put into preparation of this manuscript, I will recommend that the paper can be reconsidered, if its background is much better argued, it is substantially shortened, and it is resubmitted in a more consistent and well proof-read version.  

Author Response

Dear reviewer,

we believe that your comments on our work are very interesting and will certainly allow us to make it better, so we thank you sincerely. First of we thank you for having grasped the spirit of this work, as can be understood from the first lines of his comments and above all, for his time, for the effort used to study our manuscript carefully.

However there are some points on which we don't quite agree and now I will try to clarify our point of view.

The paper " discovers" some well known facts:

-Reaction rate is proportional to surface versus mass ratio.

-Reaction rate is proportional to concentration of H2O2.

-H2O2 is not stable at high ( boiling) temperatures.

-The oxidation of Mo is,- once initiated, extothermic, and cooling is necessary to prevent loss of H2O2 due to boiling, and,  - secondary oxidation of the Mo metal.

-A stable orange crystalline compound can be formed (sodium molybdate) by addition of sodium hydroxide and careful evaporation to dryness

None of this is hardly new. However the detailed studies and the precise measurements of temperature and reaction rate as well as the "modern methods" of x-ray diffraction, Raman and ICP-MS in the characterization of the reaction products is probable above the level found anywhere in the old literature”.

It is true that some of the points raised here may at the end seem not quite interesting and somewhat trivial after the study has been done through and we probably should have waited for these results.

However, it is not true in all the cases that the kinetics of inorganic reactions depend solely and exclusively on the specific surface of the solids.

There are cases where the chemical process follows specific paths where this is not true. For example, if the driving force is transport in the gaseous phase or eventually reaction that can occur in a different phase or if the slow stage of the process is determined by the nucleation and growth of the new phase that forms. There are many reviews that deal with the subject in a very thorough way. Just for example, we cite this one here: “Ammar Khawam and Douglas R. Flanagan, Solid-State Kinetic Models: Basics and Mathematical Fundamentals J. Phys. Chem. B 2006, 110, 17315-17328”

There are examples of reactions in which the kinetics strongly depend on the size of the particles and it is not uncommon that, for example, nanoparticles can have chemical reaction pathways that are very different from those of the materials they are made of, but in a macroscopic form. And conversely, differently activated surfaces, even of single crystals, can react very differently, even when the specific area is the same. It is also well known that, in crystals, faces with different Miller indices have chemical reactivities which can also be very different.

Analogous reasoning can also be made regarding the second point.

It is true that the concentration of the reactants (in this case in particular the concentration of hydrogen peroxide) is a fundamental parameter and that an increase often leads to a higher reaction rate. However, it is not trivial to know a priori what the law is that links the concentration of a reactant to the reaction rate. It could be linear, quadratic, logarithmic… or have an even more complex trend, especially when dealing with reactions of the radical type (which, in the case of presence of peroxides, can easily be imagined), autocatalytic or in general in heterogeneous phases.

Therefore, in general even cases that may seem "easy" often deserve to be studied in depth.

This probably goes a bit beyond the scope of this manuscript, because we were not specifically interested in studying the details of the reaction mechanism at small scales, i.e. laboratory scales, with a few milligrams of material and a few ml of liquids, but rather to see the behavior at much higher scales.

This aspect also deserves to be studied, precisely because on large scales, a perfectly known process can behave differently, due to gradients in temperature, concentration, mixing of the reactants and other factors that can only be discovered with experiments. Indeed, one of our aims was to carry out a study on a larger scale than that of a laboratory, rather than studying the precise details of microscopic the mechanism.

As regards the use of modern analysis techniques, we believe that they are very useful for determining the effectiveness of the process and one of the points to be clarified is evidently the purity of the products they can prepare.

As for the reviewer's specific points, these are our responses.

1) The justification of the paper is based on the fast neutron production of Mo-99 from kilogram quantities of metallic, natural molybdenum. True enough, such a scheme will require dissolution of the molybdenum, but NOT on the time scale of the irradiation length (line 124 etc), but on the time scale of half-life of Mo-99.  This leaves much more time for the chemical processing.

2) Even at a speculative high radionuclide yield of 2-5 Ci/24 hours, the molar activity of Mo-99 from such a production route is VERY low as compared with what is needed for any modern Mo/99/Tc-99m generator system. A typical hospital generator at 50 GBq Mo-99 would then be needed to be loaded with hundreds of grams of molybedenum.  This makes the generator itself less feasible and unpractical, and the resulting radioactive concentration of Tc-99m probably very low. Unless this problem is somehow solved ( or a solution explained) the rapid dissolution of kg's of molybdenum is not really relevant in this context.

The Sorgentina-RF project should be seen primarily as a prototype to demonstrate the feasibility of producing Tc-99 through an alternative methodology to traditional methods that rely on fission reactors. There are many reasons to look for these alternative methods, starting with making local demand independent of international production. Once the possibility of using the Sorgentina-RF machine for production has been demonstrated, it could be scaled up to even larger sizes with higher production.

But it has no ambition to be currently competitive with existing reactors, in terms of the mass of Tc or Curies produced. The estimate is that of being able to produce a few curies of activated molybdenum, starting from a few kilograms of metallic molybdenum, which however are expected to be sufficient for the needs of at least a couple of Italian regions.

It is not expected that much more Curie at the end of irradiation (EoI) can be obtained with the machine as it is currently planned, but this is not its real goal at the present. Indeed, regardless how the activation of molybdenum is carried out, this manuscript concerns only the radiochemistry part. That is to say that the authors were interested in how to complete the process of the formation of the stock solution, developing a procedure valid for non-activated materials which, however, can then be replicated also for activated targets.

For this however, it is instead very important that the dissolution process is carried out quickly: If the dissolution was too slow, the final activity would be too low to be actually useful.

Furthermore, Sorgentina-RF is designed to be able to work continuously for many days a year, except for the breaks necessary for ordinary maintenance. The determining time is the irradiation time, that is 24 hours. When a new batch arrives in the radiochemistry hot cells, the previous one has to be fully processed. Otherwise, there is also the risk that different batchs mix and that therefore the final activity of the stock solution cannot be precisely known, which is not very appreciable from an industrial point of view. On the contrary, the final concentration of the stock solution to the desired value is not realistically a time consuming problem.

Once the solution at the desired concentration is obtained, it can be parceled easily and the individual aliquots delivered (although this is beyond the scope of this manuscript)

3) While it is correct that finely powdered radioactive molybdenum will not be easy to handle in dry, open state, it can (and is) routinely and safely handled when the powder is encapsulated during irradiation and transport. More so, water soluble chemical compounds of molybdenum could be directly irradiated, making the difficult metal dissolution step unnecessary.

The possibility of irradiating other molybdenum compounds other than the metal itself or even liquid solutions had been considered. for example, if molybdenum oxide, MoO3, were irradiated, there would be much less difficulty from a chemical point of view. However, there are two main reasons that led to the exclusion of these strategies. The density of molybdenum is much higher than that of its oxides or liquid solutions of its compounds. Therefore, considering that the useful volume that will be irradiated by the neutrons is not unlimited, but instead fixed, using the metal it is possible to have a much higher irradiated mass.

Also the number of atoms per gram of molybdenum is higher in the metal than in the oxide (in MoO3, one useful atom every four). Therefore, using compounds other than the metal itself could lead to much lower yields.

Furthermore, while simulations by our specialist colleagues have shown that irradiation of molybdenum leads to the formation of very negligible amounts of unwanted secondary isotopes, this may not be the case for other compounds. The irradiation of large quantities of oxygen (in the case of oxides and of course of aqueous solutions), of chlorine, nitrogen or sulfur (if starting from chlorides, nitrates or sulphates) can lead to large quantities of unwanted radioisotopes, such as Cl-36, C-14 which are also highly mobile in the environment, thus constituting a serious radio-protection issue.

The water also acts as a shield against neutrons, making the beam less effective, both in terms of intensity and in terms of the absorption cross section by the molybdenum.

For this reasons, the choice of metal is decidedly more appropriate.

“4) The technology of rapid dissolution of radioactive metallic molybdenum (Mo-100) targets of 100-500 mg size has already been proven. The so called "Instant Technetium" cyclotron route is already using this technique, where cyclotron targets are dissolved and Tc-99m extracted on a time scale << 6 hours.  Some manufacturers prefer the dry distillation, but the wet dissolution route is also well developed.  A proper referencing to this technology ( pioneered by TRIUMF in Canada) should be made , at least

On this point we agree with the referee, however the principle of Sorgentina-RF is to want to be an innovative and different system. It is by far beyond the scope of this manuscript, but the idea is to have a small neutron source that can also be used to produce Tc-99. But in principle one could think that other radiopharmaceuticals could be obtained or that the neutrons thus created could be used for other purposes (neutron spectroscopy, sensors...) and in the complete project, in fact, other neutron extraction lines are envisaged, even at different energies. However, the reviewer's suggestion is sharp and so we will adjust our references.

These fundamental objections apart, it does not help that the manuscript itself is talkative and self-repeating in several places ( see as example lines 170-174 as compared to lines 174-177 and also lines 289-297 being exactly identical to lines 302-313) The "explanation" of the "Sorgentina RF" concept is also "explained in several places.

The reference to  table 3 in line 429 should be reference to table 2 !

The ordinate axis in fig.8b is missing ( and how can mass disappear from a chemical reaction?)    

The reviewer is absolutely right and we will correct all of these points.

The SORGENTINA RF is described as being both 300 W (line 84) and 250 kW ( line 243).  That is confusing. The description of the titanium target is a mistake. It is the H-3 that is implanted into the titanium target. This target is then bombarded with deuterons.  

Actually, one of the peculiarities of Sorgentina-RF lies precisely in the way of obtaining the nuclear fusion reaction and consequently the neutron beam. In part, this work takes place downstream of irradiation, so from a radiochemical point of view, it is not all that important to know how the target is treated. However, the accelerator was designed to accelerate both deuterium ions and tritium ions, so it's not like one of the two is implanted before the second. This is explained in our reference number 1, but in more detail in other works, already published. For instance: “Nicola Fonnesu , Salvatore Scaglione, Ivan Panov Spassovsky, Antonino Pietropaolo, Pietro Zito and The SRF Collaboration, On the definition of the deuterium-tritium ion beam parameters for the SORGENTINA-RF fusion neutron source, Eur. Phys. J. Plus (2022) 137:1150 https://doi.org/10.1140/epjp/s13360-022-03060-4”.

We can add these references to our list.

It is not justified why the all the elaborate analytical methods of XRD, Raman and ICP-MS are needed to establish the simple ( and visually obvious, from crystal shape and color)  fact that sodium molybdate has been formed. As long as there is no practical generator concept for this kind of Mo-99, the fine chemical details are unnecessary.   

As mentioned a few lines earlier, the main reason why all these analytical methods have been used is to demonstrate the effectiveness of the process. One of our intentions is to find the optimal conditions for dissolving molybdenum, with the best ratio of time, experimental conditions, amount of reagents used possible. However, for example, achieving a fast dissolution, but a low quality final product could not be considered a success. Furthermore, given that we are talking about a product that is of pharmaceutical type end use, the purity of the solutions is obviously one of the most important requirements. These analyzes were necessary to prove these points.

Indeed, the "Molecules" guidelines explicitly state that: "The correct identification of the chemical compounds is a key part of scientific information; thus, the journal will not accept papers lacking chemical characterization. For this reason, if the manuscript reports on the synthesis and/or extraction and/or characterization of chemical compounds, the authors are strongly encouraged to fill out and submit the Chemical Characterization Checklist, marking the performed analyses.”. Probably, a simple visual recognition may not be enough.

The section 3.2 about the the TEM microscope imaging of the dissolution process  leads to a speculative ( and lengthy ) explanation of the physical chemistry behind the fact that ends are dissolved at higher rate than cylinder surfaces . To this reviewer, it looks more like an artefact of the manufacturing process of the beads. They have most likely been hot drawn under high pressure and then "chopped" into short sections, leaving the grain structure aligned and closed on the cylinder surface and fractured on the ends.

We do not think that the discussion presented here is absolutely valid in every possible case and this discussion is linked to these samples and their geometry.

Despite this, we do not think we can speak of an artifact, because it is well supported by the analyzes carried out with the scanning electron microscope (SEM). Furthermore, some of the points that we propose probably have a more general validity that goes even beyond the specific case. In other words, the dissolution process at the microscopic level is certainly complex and certainly, local defects play a very important, probably decisive role. Surely some of the points we analyze depend on the production history of the samples, but this is also an absolutely general matter.

Best regards

Round 2

Reviewer 2 Report

Your manuscript about rapid dissolution of molybdenum as a step towards production of Tc-99m has been resubmitted and delegate  to me for renewed review after my first review that recommended major revision.

I have carefully read the authors comments to the reviewers ( my ) points, and we obviously disagree on the importance and novelty of the results presented in the manuscript. 

A major concern of mine is still that the claimed importance for future production of medically important Mo-99 and7or Tc-99m is still not justified:

1)The resulting specific activity ( molar activity ) is impractically low given the amounts of target molybdenum proposed.

2)The need for speed is still exaggerated. It is ultimately the half life of Mo99 that determines both the irradiation time constant and the time available for the chemistry. 

3) The irradiation of metallic molybdenum in encapsulated powder form is still not discussed in depth, although it is a clear available alternative to the presently presented method.

I have also read through the revised manuscript, and see only minor adjustments., I recommended a major revision and a shortening of the manuscript. 

I can only see minor revisions, and accordingly, I can not recommend acceptance of the paper in the present form.   

Author Response

Dear Editor,

First of all, we want to thank the reviewer for her/his time, her/his attention towards our manuscript.

We are sorry the reviewer considers this work unimportant, because for us, given the current medical importance of Tc-99m, any new advancement in the production method should be considered significant.

Here is our point-to-point response to the reviewer:

Your manuscript about rapid dissolution of molybdenum as a step towards production of Tc-99m has been resubmitted and delegate to me for renewed review after my first review that recommended major revision.

I have carefully read the authors comments to the reviewers ( my ) points, and we obviously disagree on the importance and novelty of the results presented in the manuscript. 

A major concern of mine is still that the claimed importance for future production of medically important Mo-99 and7or Tc-99m is still not justified:

1)The resulting specific activity ( molar activity ) is impractically low given the amounts of target molybdenum proposed.

2)The need for speed is still exaggerated. It is ultimately the half life of Mo99 that determines both the irradiation time constant and the time available for the chemistry. 

The supply chain of this radioisotope is vulnerable, as exhaustively demonstrated by the bibliographic references cited by us and by many others and at the same time the demand does not seem to be destined to drop in the short term. Having production capacity capable of meeting at least some local needs is important. We want to underline once more that Sorgentina - RF is a facility still in the prototype stage and does not intend to be an international or continental production center, but that once operational it will be able to satisfy the healthcare needs of some large Italian hospitals, i.e., at least an Italian region. Subsequently, the project could be extended to increase production, but this is not the current intention. This fact has been published in numerous scientific articles on many different scientific journals, in addition to those cited in the references of this manuscript, and this means that it is accepted by the international scientific community. In other words, the scientific community believes this project is capable of functioning and capable of producing Mo-99 in the expected quantities and activity.

However, according to the personal point of view of the authors, a small and local production method, but widespread in various localities, could have advantages compared to that of a single large production center, on an international scale, for example, that of securing the supply chain against a sudden interruption of the activities of that single large center. Moreover, these supply chains problems have already occurred in the past and are therefore not only hypothetical.

In the final analysis, Sorgentina-RF which is essentially only a 250kW power, accelerator-driven 14MeV fusion neutron source, will be extremely easier and cheaper to install and develop than a cyclotron or a nuclear reactor.

Nevertheless, we are aware that compared with other production methods, in particular with the production that derives from large fission plants, the production foreseen by Sorgentina RF is modest. The activity of the target is small, only a few Curies, but this stresses even more the need to do the radiochemical operations as quickly as possible, in order not to decrease even more the activity of the output stock solution. But the need of rapidity is only one of the aspects involved. In fact, the traditional production method requires the use of enormous quantities of strong acids and aggressive reagents and therefore cannot be defined as ecologically acceptable at all. Any accident at this stage would, futhermore, have extremely serious consequences.

The goal of Sorgentina - RF is also to obtain the stock solution using small quantities of reagents, eventually the smallest possible.

We are aware that our proposal is not the only possibility, but we still believe it is a valid and acceptable production method.

3) The irradiation of metallic molybdenum in encapsulated powder form is still not discussed in depth, although it is a clear available alternative to the presently presented method.

The irradiation of metallic molybdenum in encapsulated powder is a completely different production method which involves cyclotrons and protons. Our method is specifically designed for 14MeV neutrons for the irradiation of large quantities of natural molybdenum, for this reason the aforementioned method is cited but not discussed in detail.

I have also read through the revised manuscript, and see only minor adjustments., I recommended a major revision and a shortening of the manuscript. 

I can only see minor revisions, and accordingly, I can not recommend acceptance of the paper in the present form.   

In any case, the reviewer has given us the opportunity to review our work and try to improve it, hopefully in a significant way. We have, following reviewer’s advice, reduced the length of the work, eliminating everything that was superfluous or even downright unclear or confusing.

We think that the characterization of Na2MoO4 is still important, also because this material is also useful in other fields, for example in optoelectronics, but above all it is so to demonstrate that the preparation method of the mother solution is good and there are no impurities of sort that they can pollute it.

For this reason, we have modified the manuscript, not only from a formal point of view, but following a new logic.

The discussion section has been divided into three subsections, while before there were only two, because we have separated the preparation method and the description of the optimal conditions for the preparation of the stock solution from the rest. We believe that the microscopic description is still important, even from a chemical point of view, to better understand the underlying mechanism and finally we have shown that the product that can be obtained is of excellent quality.

We also followed the advices that were given to us along the rejection with encourage resubmission decision by the Editor. They were mainly about the shortening of the manuscript and the removal of repetitions. We sharpened some of the formulations and worked on the consistency of the arguments.

Now, having made these changes, we hope that our manuscript is improved in quality and can be judged good for publication.

Best regards with the trust of all the authors,

Alberto Ubaldini
